# Halenaquinol Blocks Staphylococcal Protein A Anchoring on Cell Wall Surface by Inhibiting Sortase A in *Staphylococcus aureus*

**DOI:** 10.3390/md22060266

**Published:** 2024-06-10

**Authors:** Jaepil Lee, Jae-Hyeong Choi, Jayho Lee, Eunji Cho, Yeon-Ju Lee, Hyi-Seung Lee, Ki-Bong Oh

**Affiliations:** 1Department of Agricultural Biotechnology, College of Agriculture and Life Sciences and Natural Products Research Institute, Seoul National University, Seoul 08826, Republic of Korea; ljp0926@snu.ac.kr (J.L.); jayho@snu.ac.kr (J.L.); eunji525@snu.ac.kr (E.C.); 2Marine Natural Products Chemistry Laboratory, Korea Institute of Ocean Science and Technology, Busan 49111, Republic of Korea; jhchoi@azcuris.com (J.-H.C.); yjlee@kiost.ac.kr (Y.-J.L.); 3Department of Applied Ocean Science, University of Science and Technology, Daejeon 34113, Republic of Korea

**Keywords:** *Staphylococcus aureus*, sortase A, staphylococcal protein A, marine sponge, *Xestospongia* sp., halenaquinol

## Abstract

Sortase A (SrtA) is a cysteine transpeptidase that binds to the periplasmic membrane and plays a crucial role in attaching surface proteins, including staphylococcal protein A (SpA), to the peptidoglycan cell wall. Six pentacyclic polyketides (**1**–**6**) were isolated from the marine sponge *Xestospongia* sp., and their structures were elucidated using spectroscopic techniques and by comparing them to previously reported data. Among them, halenaquinol (**2**) was found to be the most potent SrtA inhibitor, with an IC_50_ of 13.94 μM (4.66 μg/mL). Semi-quantitative reverse transcription PCR data suggest that halenaquinol does not inhibit the transcription of *srtA* and *spA*, while Western blot analysis and immunofluorescence microscopy images suggest that it blocks the cell wall surface anchoring of SpA by inhibiting the activity of SrtA. The onset and magnitude of the inhibition of SpA anchoring on the cell wall surface in *S. aureus* that has been treated with halenaquinol at a value 8× that of the IC_50_ of SrtA are comparable to those for an *srtA*-deletion mutant. These findings contribute to the understanding of the mechanism by which marine-derived pentacyclic polyketides inhibit SrtA, highlighting their potential as anti-infective agents targeting *S. aureus* virulence.

## 1. Introduction

The antibiotic resistance of important Gram-positive pathogens that commonly cause serious infections is a significant global health concern [1]. In response to the escalating threat of antimicrobial resistance, the proposal of anti-infective agents emerges as a vital strategy. While traditional antimicrobial treatments target the elimination or inhibition of microorganisms, thereby fueling selective resistance pressures, anti-infective agents take a distinct approach by impeding pathogenicity through the inhibition of virulence factors [2]. These virulence factors, utilized by microbial pathogens, play multifaceted roles in bacterial adherence, biofilm formation, quorum sensing systems, the secretion of toxins, and the orchestration of two-component systems. Gram-positive bacteria, in particular, wield an arsenal of surface proteins that function as virulence factors to facilitate pathogenicity and infection. These proteins play crucial roles in adhering to and invading host endothelial tissues while evading host complement proteins and immunoglobulins. Sortase A (SrtA) inhibitors exemplify this approach [3]. SrtA was first discovered in *S. aureus* by Schneewind and colleagues, and the SrtA-catalyzed sorting reaction of surface proteins in *S. aureus* is well understood [4].

Gram-positive bacteria, in particular, employ various surface proteins as virulence factors to mediate pathogenicity and infectivity [5,6,7]. Despite their diverse functions, these surface proteins share a common C-terminal LPXTG motif, acting as a cell wall sorting signal. SrtA, a cysteine transpeptidase that is bound to the periplasmic membrane [8] plays a crucial role in attaching these surface proteins to the peptidoglycan cell wall. The virulence factors of *S. aureus* encompass cell-surface proteins, secretory toxins, and various exoenzymes that contribute to bacterial adhesion, tissue invasion, and the evasion of host defense [9]. The cell surface displays more than 20 proteins that are covalently attached to peptidoglycan, with catalysis by SrtA. For instance, staphylococcal protein A (SpA) interacts with the Fc region of immunoglobulins, hindering opsonophagocytosis upon entering the human host [10]; clumping factors A and B (ClfA and ClfB) function to adhere to immobilized fibrinogen for immune evasion; and fibronectin-binding proteins A and B (FnBPA and FnBPB) adhere to the extracellular matrix [11]. The disruption of SrtA’s function, as observed in *S. aureus srtA*-knockout mutants, leads to a failure in displaying surface proteins with the LPXTG motif, impairing infection in mouse models [12]. Inhibiting SrtA could effectively disarm multiple virulence mechanisms concurrently, rendering it a promising approach to neutralize pathogens that are reliant on SrtA for virulence. Due to the pivotal role of bacterial adhesion that is mediated by surface proteins in Gram-positive bacterial pathogenesis, SrtA emerges as an attractive target for anti-infective agents.

In drug discovery, the importance of natural products sourced from marine organisms has been widely acknowledged and emphasized in the various literature [13,14,15]. The deep-sea environment offers unique growth conditions, including minimal light, high pressure, salinity, extreme temperatures, and oxygen deprivation [16,17]. These conditions may confer distinctive characteristics on marine organisms, potentially yielding novel bioactive secondary metabolites compared to their terrestrial counterparts [18]. During the exploration of marine organisms for SrtA inhibitory metabolites, we encountered the sponge *Xestospongia* sp., collected from Bohol Island in the Philippines. A chemical analysis of this organism resulted in the discovery of halenaquinol (**2**) and five related pentacyclic polyketides. This study explores the structures and inhibitory effects of these compounds on SrtA. Notably, halenaquinol (**2**) showed a significant inhibition of *S. aureus*-derived SrtA, and its in vivo effectiveness and mode of action were linked to the suppression of the SrtA-mediated cell wall surface display of SpA.

## 2. Results

### 2.1. Structural Elucidation of Compounds ***1**–**6***

The crude extract of *Xestospongia* sp. contained several polyketides that possess a polycyclic ring system. Using combined spectroscopic analyses and a comparison of NMR data with those previously reported [19], compound **1** was identified as a halenaquinol sulfate. The ^1^H and ^13^C NMR spectra of compounds **1**–**6** were closely related to each other, and all compounds possessed a pentacyclic moiety. By comparing the spectral data of these compounds with previously reported spectral data (Appendix A), their structures were readily identified as being those of a halenaquinol (**2**) [19], 13,14,15,16-tetrahydroxestoquinol (**3**) [20], xestoquinol sulfate (**4**) [21], secoadociaquinone B (**5**) [22], and xestolactone A (**6**) [23] (Figure 1).

### 2.2. SrtA Inhibitory Activities of Compounds ***1**–**6***

The SrtA inhibitory activity of compounds **1**–**6**, along with that of a known SrtA inhibitor, quercetin [24], was evaluated in vitro using a fluorescence-based enzymatic assay. Recombinant SrtA from *S. aureus* ATCC6538p was expressed in *Escherichia coli* and was purified using nickel affinity chromatography [25]. The cleavage of a synthetic peptide substrate by the SrtA enzyme emitted fluorescence, which was then measured [26]. Table 1 presents the half maximal inhibitory concentration (IC_50_) values calculated from SrtA inhibition rates. Quercetin, a positive control, showed moderate SrtA inhibitory activity, with an IC_50_ value of 157.62 μM. Halenaquinol, 13,14,15,16-tetrahydroxestoquinol, and xestolactone A were identified as strong inhibitors, with IC_50_ values of 13.94 μM, 51.76 μM, and 47.52 μM, respectively. Halenaquinol sulfate and secoadociaquinone B exhibited moderate inhibitory activities, similar to that of quercetin. Notably, halenaquinol showed the strongest inhibitory activity (IC_50_ = 13.94 μM), approximately 9.2 times more potent than halenaquinol sulfate (IC_50_ = 103.48 μM). These results suggest the importance of the hydroxyl group at the C-16 position of these compounds for SrtA inhibitory activity. In addition, 13,14,15,16-Tetrahydroxestoquinol showed potent inhibitory activity (IC_50_ = 51.76 μM), whereas xestoquinol sulfate completely lost its inhibitory activity (IC_50_ > 319.69 μM). In contrast, xestolactone A, which has a hydroxyl group at the C-3 position, showed strong SrtA inhibitory activity (IC_50_ = 47.52 μM), indicating the crucial role of the carbonyl group or the hydroxyl group at the C-3 position in the SrtA activity inhibition of these compounds. As SrtA inhibitors act as anti-infective agents by disrupting bacterial pathogenesis without affecting microbial viability [6], minimum inhibitory concentration (MIC) values were examined to determine their effects on bacterial cell growth. All compounds showed no inhibitory activity on the cell growth of *S. aureus* (MIC > 128 μg/mL) (Table 1).

To determine the type of inhibition, kinetic studies were performed with halenaquinol at IC_50_ or twofold IC_50_ based on a Lineweaver and Burk plot (Figure 2A) [27]. Inhibitor constants were obtained by a Dixon plot (Figure 2B) [28]. Inhibitory kinetics show that halenaquinol behaved as a mixed inhibitor (*K*i = 9.55 µM) (Figure 2). Moreover, the binding of halenaquinol to SrtA was reversible because the enzyme activity was recovered by dialysis, excluding the possible existence of a covalent bond between inhibitor and enzyme.

### 2.3. Effect of Halenaquinol on the Transcription Levels of srtA

To ascertain whether halenaquinol inhibits the transcription of *srtA* or blocks its function at an enzyme level without affecting the expression of the *srtA* gene, the mRNA expression level of *srtA* in response to various concentrations and exposure times of halenaquinol was initially explored using semi-quantitative reverse transcription PCR (RT-PCR). Total RNA isolated from *S. aureus* Newman was transcribed into cDNA, and it was used as a template to amplify the target genes [29]. A time-course analysis assessed the impact of halenaquinol on *srtA* expression at different intervals from 0 to 60 min. The PCR band intensity representing *srtA* mRNA expression increased over time, whereas the band intensity for the positive control, *gyrB* (encoding for the B subunit of the DNA gyrase)–a housekeeping gene in *S. aureus*–remained consistent (Figure 3A). Furthermore, the expression level of *srtA* was analyzed after 40 min of incubation at various concentrations of halenaquinol. The PCR band intensity representing the mRNA expression of *srtA* and *spa* slightly increased with the increasing concentrations of halenaquinol (Figure 3B). These findings suggest that halenaquinol does not inhibit the transcription of *srtA* and *spA*.

### 2.4. Halenaquinol Blocks SpA Anchoring on Cell Wall Surface by Inhibiting SrtA

As indicated in Table 1, the effects of halenaquinol on SrtA activity in vitro were investigated using recombinant SrtA that was derived from *S. aureus* strain ATCC6538p. SpA of *S. aureus* was secreted as a precursor with the LPXTG motif that was cleaved by SrtA [30]. Therefore, the effect of halenaquinol on the in vivo inhibition of *S. aureus* SrtA was investigated by quantifying the abundance of SpA. This experiment involved *S. aureus* Newman and its isogenic *srtA*-knockout mutant. The *srtA* sequence (621 base pairs) of the Newman strain (GenBank accession number: AP009351) showed a 99% similarity with that of the ATCC6538p strain (GenBank accession number: CP041746). 

The anchoring of SpA on the cell-wall surface in *S. aureus* was measured using Western blot analysis with monoclonal anti-SpA antibody (Figure 4). First, to detect the abundance of newly displayed proteins after treatment with the test compound, cell-surface proteins, including SpA, were removed using trypsin (Figure 4A) [31]. In the wild type (WT), cell wall surface-displayed SpA was removed using trypsin treatment for 1 h at 37 °C, comparable to the level in the *srtA*-knockout mutant (*ΔsrtA*). In trypsin-treated staphylococci incubated for 1 h without trypsin, wild-type *S. aureus* deposited newly synthesized SpA on the cell wall surface, which was not observed in the *srtA*-knockout mutant (Figure 4B). Then, trypsin-treated wild-type *S. aureus* cells were incubated in medium containing an SrtA inhibitor, halenaquinol, over a concentration range of 1-to-4 times the IC_50_ values. The anchoring of SpA in the treatment with 1 × IC_50_ (13.94 μM) showed no significant difference from the wild type. However, its anchoring was significantly reduced in treatments with 2 × IC_50_ (27.88 μM) and 4 × IC_50_ (55.76 μM), showing reductions of approximately 53% and 85%, respectively (Figure 4B,C). Similar results were obtained with quercetin, a control compound. 

Upon staining *S. aureus* cells with a fluorescein isothiocyanate (FITC) antibody [31,32] and immunofluorescence microscopy images revealed the blocking of SpA surface display through SrtA inhibition. In the absence of halenaquinol, SpA was abundantly displayed on wild-type *S. aureus*, whereas surface display was significantly reduced in the presence of halenaquinol at a concentration of 8 × IC_50_ (111.52 μM) (Figure 5). These results demonstrate that halenaquinol blocks the cell-wall surface anchoring of SpA by inhibiting SrtA, thereby confirming its role as an effective SrtA inhibitor.

## 3. Discussion

The fused furan polycyclic core of pentacyclic polyketides is present in natural products and demonstrates notable biological activities [33]. For instance, halenaquinol and halenaquinone are five-ring polyketides isolated from marine sponges and have shown potential as antibiotics, cardiotonics, and inhibitors of protein tyrosine kinases. This study revealed the SrtA inhibitory activities of six pentacyclic polyketides isolated from *Xestospongia* sp. Also, the structures of these compounds were elucidated using spectroscopic techniques and compared with previously reported data. Among them, halenaquinol was identified as the most potent SrtA inhibitor. The SrtA inhibitory activity data of the isolated compounds indicated the critical role of the hydroxyl group at the C-16 position and the carbonyl group or the hydroxyl group at the C-3 position in SrtA inhibitory activity (Figure 1). Semi-quantitative RT-PCR data, Western blot analysis, and immunofluorescence microscopy images suggest that halenaquinol blocks the cell wall surface anchoring of SpA by inhibiting the activity of SrtA, rather than inhibiting the transcription of *srtA*, highlighting its potential as an anti-infective agent targeting *S. aureus* virulence. 

Over the past decade, an impressive array of marine natural products featuring novel structures and a wide range of bioactivities, including anti-SrtA activity, have been identified [34,35]. Notably, marine sponge-derived natural products such as topsentin and hamacanthin classes (bromotopsentin, IC_50_ = 39.6 μM), psammaplin A (IC_50_ = 47.4 μM), and aaptamines (isoaaptamine, IC_50_ = 16.2 μM) showed potent inhibitory activities against *S. aureus* SrtA [36,37,38]. The scaffold of topsentin A was used as a model ligand to identify 3-(4-pyridinyl)-6-(2-sodiumsulfonatephenyl)[1,2,4]triazolo [3,4-b][1,3,4]thiadiazole and related compounds through virtual screening and optimization (compound **6**e, IC_50_ = 9.3 μM), which block *S. aureus* SrtA activity in vitro and in vivo [26]. Among the diarylacrylonitriles synthesized and tested for SrtA inhibition, (*Z*)-3-(2,5-dimethoxyphenyl)-2-(4-methoxyphenyl) acrylonitrile was the most active compound with an IC_50_ value of 9.2 μM, behaving as a competitive inhibitor and binding the enzyme in a reversible manner [25]. Many natural products derived from medicinal plants have been investigated as inhibitors of SrtA on the basis of its functions. For example, using a fluorescence resonance energy transfer assay and molecular docking, myricetin, palmatine, andesculetin were screened from 56 compounds and found to inhibit SrtA activity, with IC_50_ values ranging from 4.63 to 52.84 μM [39]. Compared to these compounds, halenaquinol, derived from *Xestospongia* sp., with an IC_50_ of 13.94 μM, is believed to possess an exceptionally strong SrtA inhibitory activity and is, therefore, expected to function as an anti-infective agent, potentially disrupting bacterial infection mechanisms without adversely affecting microbial viability.

SrtA inhibitors aim to solely inhibit enzyme activity and do not affect transcription at the gene level. As depicted in RT-PCR data (Figure 3), the intensity of PCR band expression increased in mRNA expression for the *srtA* and *spa* genes after treatment with halenaquinol at distinct concentrations and time points. However, the housekeeping gene *gyrB* remained consistently expressed. *S. aureus* operates various virulence regulatory systems; for example, the *agr* quorum-sensing system, cytoplasmic SarA-family regulators, and others [40]. The accessory gene regulator (*agr*) is the most studied system that encodes the quorum-sensing circuit, sensing the cell’s population density and regulating gene expression patterns. As the population reaches a threshold, it upregulates secreted protein production and downregulates cell wall-associated factors, including surface proteins [41,42]. Considering the presence of complex interconnected networks within *S. aureus*, the increase in mRNA could be presumed to result from an unknown negative feedback loop. The SrtA inhibitor blocks the transpeptidation of SrtA, which, in turn, fails to anchor surface proteins to the cell wall; consequently, cells without surface proteins become unable to utilize various modes of action. Cells may send signals that lead to an increase in srtA mRNA expression level. It is hypothesized that this unknown negative feedback loop repeats, resulting in the upregulation of *srtA* and *spa* mRNA expression.

A similar case was reported in *Streptococcus mutans* [43], whereby the upregulation of the mRNA expression of *srtA* and *pac* in *S. mutans* was observed under myricetin treatment at different concentrations; myricetin exhibited an inhibitory activity of *S. mutans* SrtA (IC_50_ = 15.48 μg/mL) and the surface protein Pac in *S. mutans* is catalyzed by SrtA. The homology in the amino acid sequence between *S. aureus* and *S. mutans* SrtA is high (43%), and their sorting mechanisms are quite similar [44]. This observation explained a consistent decrease in Pac levels within cells and an increase in Pac levels in the supernatant, correlating with the increased mRNA expression of *pac* and *srtA* [43]. In this study, even though *srtA* and *spa* mRNA expression was increasing, treatment with halenaquinol (**2**) resulted in the inhibition of SrtA activity, evidenced by its decreased abundance in Western blot analysis and immunofluorescence microscopy images (Figure 4B,C and Figure 5). Taken together, the increase in mRNA expression may occur through unknown pathways and halenaquinol could be considered to inhibit SrtA not at the gene level, but at the enzyme level.

While our findings demonstrate the potential of marine sponge *Xestospongia* sp.-derived halenaquinol and related pentacyclic polyketides as SrtA inhibitors, more experiments are needed to validate their efficacy as anti-infective agents. This includes in vitro studies to assess their effects on bacterial adherence and biofilm formation, as well as in vivo studies using infection models to evaluate their therapeutic effects. Additionally, employing molecular modeling techniques can provide insights into the interaction between these compounds and SrtA, guiding further drug development efforts.

## 4. Materials and Methods

### 4.1. Animal Material

Specimens of *Xestospongia* sp. sponge were collected by hand using scuba gear at a depth of 20–35 m off the shore of Bohol Island in the Philippines, in March 2018. The marine sponge *Xestospongia* sp. was identified by Dr. Young-A Kim, Hannam University in Daejeon, Republic of Korea. A voucher specimen (sample no. 183PIL-101) was deposited at the department of marine bioresources, Korea Institute of Ocean Science and Technology.

### 4.2. Extraction and Isolation

The collected marine sponge *Xestospongia* sp. (1.15 kg) was kept frozen at −20 °C until it was chemically analyzed. The specimens were lyophilized and then subjected to extraction with methanol (2 L × 2) and dichloromethane (1 L × 1). The combined extracts (78.6 g) were partitioned between water and *n*-butanol (29.7 g); the latter fraction was re-partitioned using 15% aqueous methanol (14.9 g) and *n*-hexane (15.2 g). The residue from the aqueous methanol layer was subjected to C_18_ reversed-phase flash chromatography using gradient mixtures of methanol and water (elution order: 60%, 50%, 40%, 30%, 20%, 10% aq. methanol, and 100% methanol). The 60% aqueous methanol fraction (4.37 g) was further purified using reversed-phase HPLC (YMC-Pack Pro C_18_ columns, 250 × 10 mm; 65% aqueous methanol), with the major products being eluted in the following order: **4**, **1**, **2**, **3**, **6**, and **5**. Proton NMR analysis indicated that compound **1** was pure, while the others showed impurities. Further purification of these compounds was performed using reversed-phase HPLC (YMC-ODS-A C_18_ column, 250 × 4.6 mm; 60% aqueous methanol with 0.1% trifluoroacetic acid), leading to isolated yields of 59.2, 11.5, 1.2, 0.4, 2.6, and 0.7 mg for compounds **1**–**6**, respectively. NMR spectra were acquired on a Bruker 600 MHz spectrometer at 297.1 K, with samples in CD_3_OD or DMSO-*d*_6_. High-resolution mass spectra were obtained with a Waters Q-TOF spectrometer and a JEOL JMS-700 spectrometer, both equipped with different ionization sources, ESI and FAB, respectively.

### 4.3. SrtA Inhibition Assay

The recombinant SrtA protein from *S. aureus* ATCC6538p was prepared using a previously described method [25]. Each reaction comprised a 100 μL mixture containing a reaction buffer (50 mM Tris-HCl pH 7.5, 150 mM NaCl, and 5 mM CaCl_2_), 250 ng of synthetic fluorescent peptide substrate (Abz-LPETG-Dnp) [26], 20 μg of SrtA, and varying concentrations of test samples. DMSO served as the negative control, while quercetin [24] was employed as the positive control. Test sample was dissolved in DMSO and then diluted with reaction buffer to yield a final DMSO concentration of 1%. The reaction mixture was aliquoted into a 96-well plate. Following the addition of SrtA, fluorescence intensity was determined using a microplate reader at excitation and emission wavelengths of 320 nm and 420 nm, respectively. The plate was then incubated for 1 h at 37 °C, and fluorescence intensity was measured. The effect of the inhibitor on SrtA was calculated as a percentage relative to the DMSO-treated control. The percentage of SrtA inhibition was calculated using the following equation: {1 − (Difference value of the test compound/Difference value of a negative control)} × 100. A 12-point, twofold serial dilution dose–response assay was performed independently in triplicate. The resultant dose–response concentration range was 128 to 0.1 μg/mL of inhibitor. Data were scaled to internal controls, and a four-parameter logistic model (SigmaPlot ver. 10.0) was used to fit the measured data and determine the IC_50_ values. In order to investigate the inhibition type and inhibition constant (*K*i) values of halenaquinol, SrtA enzyme activity was monitored in various concentrations of substrate in the absence and presence of halenaquinol. All reactions were repeated three times and enzyme kinetics were evaluated using Lineweaver–Burk [27] and Dixon plots [28].

### 4.4. Antibacterial Activity

An assessment of the antibacterial properties of the isolated compounds was conducted in accordance with the protocols provided by the Clinical and Laboratory Standard Institute (CLSI) [45]. The *S. aureus* ATCC6538p strain was cultivated overnight at 37 °C in Mueller–Hinton broth (MHB, BD Difco, Sparks, MD, USA). Subsequently, it was harvested using centrifugation, rinsed with sterilized deionized water, and adjusted to the optical density of a 0.5 McFarland standard (625 nm). Test compounds, initially dissolved in DMSO, were further diluted with MHB to achieve a range of 2-fold serial dilutions from 0.25 to 128 μg/mL (maintaining a final DMSO concentration of 1%). In a 96-well plate, 90 μL of the MHB containing the test compound were combined with 10 μL of broth inoculated with the bacterium (final bacterial concentration: 5 × 10^5^ cfu/mL). Following a 12 h incubation at 37 °C, the MICs were determined as the lowest concentration at which bacterial growth was inhibited. Ampicillin (Duchefa, Amsterdam, The Netherlands) served as the reference standard.

### 4.5. Semiquantitative Reverse Transcription PCR Analysis

The *S. aureus* Newman strain was inoculated into 5 mL of trypticase soy broth (TSB) and incubated overnight at 37 °C with shaking. Cultures were diluted 1:100 into TSB and grown until the OD_600_ reached approximately 0.3. Cultures (5 mL) were aliquoted, and the test compound was added with less than 0.2% DMSO. The negative control group was treated with DMSO only. Cultures were then incubated at 37 °C with shaking and harvested at specific incubation times. Then, 1 mL of each culture was centrifuged, and the pellet was stored at −80 °C. RNA was isolated using the easy-BLUE total RNA extraction kit (iNtRON Biotechnology, Boston, MA, USA). The bacterial pellets were mixed with 1 mL of RNA extraction buffer and were mechanically disrupted using a homogenizer with 0.2 g of acid-washed glass beads (≤106 μm, Sigma-Aldrich, Saint Louis, MO, USA). Total RNA was treated with DNase I and reverse transcribed into cDNA using the SuperiorScript III cDNA synthesis kit (Enzynomics, Daejeon, Republic of Korea). Amplification was conducted using the following primers: *gyrB* (5′-ATGAAAATCCACAAGTCGCA-3′ and 5′-TCTAACGCTGATTTACGACG-3′), *srtA* (5′-CTGGTGTGGTACTTATCCTA-3′ and 5′-GAATTTGAGGTTTAGCTTGC-3′), and *spa* (5′-CTTGAGCTTTGTTAGCATCT-3′ and 5′-GAGTACATGTCGTTAAACCT-3′), under the following conditions: an initial incubation at 95 °C for 2 min; followed by 30 cycles of 30 s at 95 °C, 30 s at 50 °C, and 30 s at 72 °C; and, finally at 72 °C for 5 min. The *gyrB* gene was used as a loading control.

### 4.6. Re-expression of the Cell Surface Protein

The effect of the test compound on the in vivo inhibition of *S. aureus* SrtA was investigated by quantifying the abundance of SpA, catalyzed by the transpeptidation of SrtA on the bacterial surface [26]. To detect the abundance of newly displayed proteins after treatment with the test compound, cell-surface proteins, including SpA, were removed using trypsin [31]. For this study, *S. aureus* Newman (wild-type), a human clinical isolate, and its isogenic *srtA*-deletion mutant (*ΔsrtA*) were utilized [6,37]. These strains were inoculated into 5 mL of TSB and incubated for 16 h at 37 °C with shaking. Cultures were then diluted 1:100 into TSB and were grown until the OD_600_ reached approximately 0.5. Cells from a 10 mL culture were harvested, rinsed with phosphate-buffered saline (PBS, Sigma–Aldrich), suspended in 5 mL of PBS containing 0.2 mg/mL trypsin (Sigma-Aldrich), incubated at 37 °C with shaking for 1 h, and washed with PBS. Subsequently, the cells were incubated in 5 mL of TSB containing 1 mM phenylmethylsulfonyl fluoride (PMSF), with the test compound at 37 °C with shaking for 1 h. Negative control groups were treated with DMSO solvent instead of the test compound. Then, aliquots (1 mL) were removed, and the cells were harvested and stored at −80 °C.

### 4.7. Labeling and Isolation of the Cell Surface Protein

Biotinylation was employed for labeling surface-displayed proteins, followed by their enrichment through affinity purification. The isolation procedure was adapted from a previously described method [46]. *S. aureus* cells, prepared as described above, were suspended in 1 mL of ice-cold PBS containing 1 mM PMSF. A 1% (*w*/*v*) solution of Sulfo-NHS-SS-Biotin was prepared, and it was added (100 μL) to 1 mL of cells, resulting in a final concentration of 1.5 mM Sulfo-NHS-SS-Biotin. The cells were then incubated with gentle shaking at 80 rpm for 2 h on ice and were collected using centrifugation (4000× *g*, 5 min, 4 °C). To halt the biotinylation reaction, cells were washed three times with 1 mL of ice-cold PBS containing 500 mM glycine. Next, the cells were suspended in 500 μL of lysis buffer containing 100 μg/mL lysostaphin (Sigma–Aldrich) and 2 mM PMSF. After 1 h incubation at 37 °C, the supernatants containing cell surface proteins were carefully collected using centrifugation (7000× *g*, 20 min, 4 °C). The column was loaded with 100 μL of NeutrAvidin Agarose (Thermo Fisher Scientific, Waltham, MA, USA) slurry and washed twice with PBS containing 1% Triton X-100, using centrifugation (1200× *g*, 1 min). The supernatants were then added to the column and incubated with gentle shaking at 80 rpm for 90 min on ice. Subsequently, the column was washed six times with PBS containing 1% Triton X-100 and was eluted while incubating in SDS sample buffer at room temperature for 1 h. The eluted protein samples were aliquoted and stored at −80 °C.

### 4.8. Western Blot Analysis

Protein samples were resolved on a 10% polyacrylamide gel through SDS-PAGE and were transferred to a polyvinylidene difluoride (PVDF) membrane (0.45 μm; Cytiva, Marlborough, MA, USA) with transfer buffer (25 mM Tris-HCl, 190 mM glycine, 20% methanol) at 100 V for 1 h on ice. The membrane was blocked in a blocking buffer (50 mM Tris-HCl pH 7.5, 150 mM NaCl, 3% BSA, and 0.1% Tween 20) at 25 °C for 1 h. For antibody incubations, a monoclonal anti-SpA antibody (Sigma-Aldrich) was diluted at a ratio of 1:5000 in the blocking buffer and applied as the primary antibody. The secondary antibody, a polyclonal anti-mouse IgG conjugated to alkaline phosphatase antibody (Sigma–Aldrich), was diluted at the same ratio. Membranes were incubated in the primary antibody solution at 25 °C for 1 h, washed twice for 10 min each in TBST (50 mM Tris-HCl pH 7.5, 150 mM NaCl, and 0.1% Tween 20), incubated in the secondary antibody solution at 25 °C for 1 h, washed twice, and equilibrated for 5 min in a detection buffer (0.1 M Tris-HCl, 0.1 M NaCl pH 9.5). CDP-Star (Roche, Basel, Switzerland), a chemiluminescent substrate for alkaline phosphatase, was applied to the blot, and the signal intensity was detected using a chemiluminescence imager (Vü-C, Pop-Bio Imaging, Cambridge, UK). The quantification of bands was performed using ImageJ software (NIH, Bethesda, MD, USA) and statistical analysis was conducted utilizing GraphPad Prism software (Graphpad, San Diego, CA, USA).

### 4.9. Immunofluorescence Microscopy

The surface display of SpA was visualized following established procedures [31,32]. A solution of 0.01% poly-L-lysine (Sigma-Aldrich) was added to each well of an 8-well glass chamber slide (SPL) and incubated at 25 °C for 5 min to coat the slide. After washing each well with distilled water and ensuring complete drying, 1 mL of cells, prepared as described above for the re-expression of the cell surface protein, was suspended in PBS containing 0.05% Tween 80 (PBST) and was thoroughly vortexed. A total of 250 μL of suspension was removed and incubated at 25 °C for 20 min with 250 μL of PBS (2.4% paraformaldehyde, 0.006% glutaraldehyde, and 0.05% Tween 80). The fixed cells were washed twice with PBST, and 150 μL of cells were added to the coated slide and incubated at 25 °C for 5 min. Each well was washed three times and blocked in a blocking buffer (3% BSA, 50 mM Tris-HCl pH 7.5, 150 mM NaCl, and 0.1% Tween 20) at 25 °C for 45 min. The primary antibody was diluted at a ratio of 1:1000 in the blocking buffer, while the secondary antibody was diluted at a ratio of 1:100. Wells were incubated overnight at 4 °C in the primary antibody solution, washed seven times in PBST, and incubated in the secondary antibody solution in the dark at 25 °C for 1 h. After washing seven times, anti-fade fluorescence mounting medium (Abcam, Cambridge, UK) was added. Cells were visualized using LSM 980 confocal microscopy (Carl Zeiss, Oberkochen, Germany) with an airy scan under a 63×/1.40 oil objective. Images were captured using AiryScan fluorescence detectors and a transmitted light detector (T-PMT) under 10× scan zoom, bidirectional scan, 4× averaging. All the images were analyzed in ZEN 2.1 software (Carl Zeiss).

### 4.10. Statistical Analysis

Statistical analysis was conducted using GraphPad Prism software (Graphpad, San Diego, CA, USA). Statistical significance was determined with unpaired, two-tailed Student’s *t*-test, and a *p*-value < 0.05 was accepted as being statistically significant.

## 5. Conclusions

This study explored the SrtA inhibitory activities of pentacyclic polyketides isolated from the marine sponge *Xestospongia* sp. The chemical analysis of this organism led to the isolation of halenaquinol and five related pentacyclic polyketides. The anti-SrtA activities of these compounds were also identified, with varying degrees of potency. Among them, halenaquinol significantly inhibited *S. aureus*-derived SrtA with an IC_50_ of 13.94 μM (4.66 μg/mL). The SrtA inhibitory activity data of the isolated compounds indicated the critical roles of the hydroxyl group at the C-16 position and the carbonyl group or the hydroxyl group at the C-3 position in SrtA inhibitory activity. The efficacy of halenaquinol in blocking bacterial virulence through the inhibition of the SrtA enzyme was demonstrated. Semi-quantitative reverse transcription PCR data suggest that halenaquinol does not inhibit the transcription of *srtA* and *spA*, while Western blot analysis and immunofluorescence microscopy images suggest that it blocks the cell wall surface anchoring of SpA by inhibiting the activity of SrtA, highlighting its potential as an anti-infective agent targeting *S. aureus* virulence. Further in vitro and in vivo studies are required to confirm the therapeutic potential of these compounds as anti-infective agents, with molecular modeling potentially guiding future drug development.

## Figures and Tables

**Figure 1 marinedrugs-22-00266-f001:**
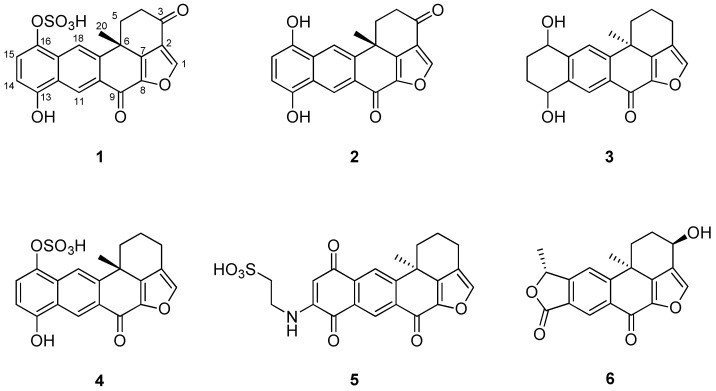
Structures of compounds **1**–**6** isolated from *Xestospongia* sp.: halenaquinol sufate (**1**), halenaquinol (**2**), 13,14,15,16-tetrahydroxestoquinol (**3**), xestoquinol sulfate (**4**), secoadociaquinone B (**5**), and xestolactone A (**6**).

**Figure 2 marinedrugs-22-00266-f002:**
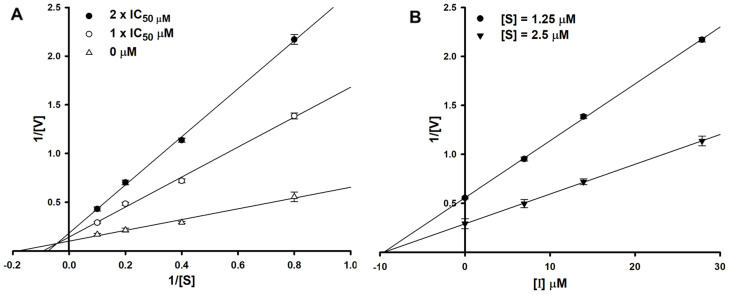
Lineweaver–Burk plot (**A**) and Dixon plot (**B**) of SrtA inhibition by halenaquinol. [S], substrate concentration (μM); [V], reaction velocity (Δabsorbance unit/min); [I], inhibitor concentration (μM). Each data point represents the mean of three experiments.

**Figure 3 marinedrugs-22-00266-f003:**
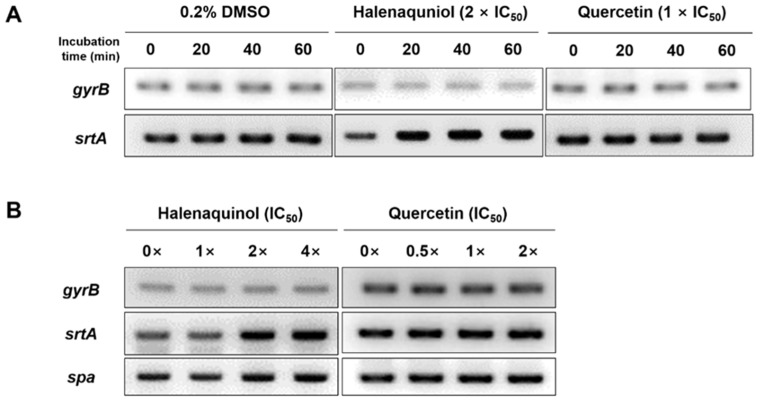
Effect of SrtA inhibitors on *srtA* and *spa* mRNA expression. *S. aureus* Newman cells were incubated in the presence of halenaquinol and quercetin. mRNA expression levels of cells were examined using semiquantitative RT-PCR. (**A**) The time courses of mRNA expression of *gyrB* and *srtA* were monitored. Total RNA was isolated from cells grown in medium containing 0.2% dimethyl sulfoxide (DMSO, negative control), halenaquinol or quercetin (positive control). (**B**) Cells were grown in medium containing the indicated concentrations of halenaquinol or quercetin. The *gyrB* gene was used as a loading control.

**Figure 4 marinedrugs-22-00266-f004:**
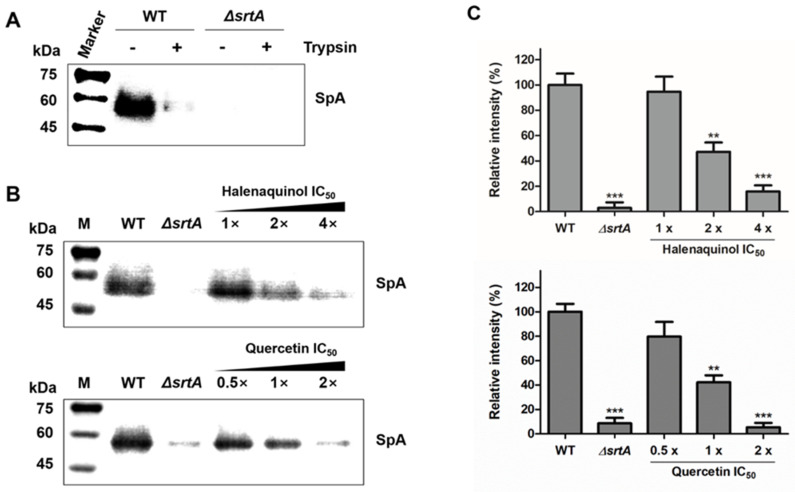
Effects of SrtA inhibitors on *S. aureus* cell wall anchoring of SpA. (**A**) To investigate the newly displayed SpA, the cell wall-anchored SpA of *S. aureus* Newman (WT, wild type) and its isogenic *srtA*-knockout mutants (*ΔsrtA*) were removed by cleaving cell surface proteins with trypsin. (**B**) The anchoring of SpA by SrtA was inhibited by halenaquinol and quercetin (control) with the indicated SrtA IC_50_ values. (**C**) Data are presented as the mean ± standard deviation of three independent experiments. Statistical significance (** *p* <0.01, *** *p* < 0.005) was determined with an unpaired, two-tailed Student’s *t*-test.

**Figure 5 marinedrugs-22-00266-f005:**
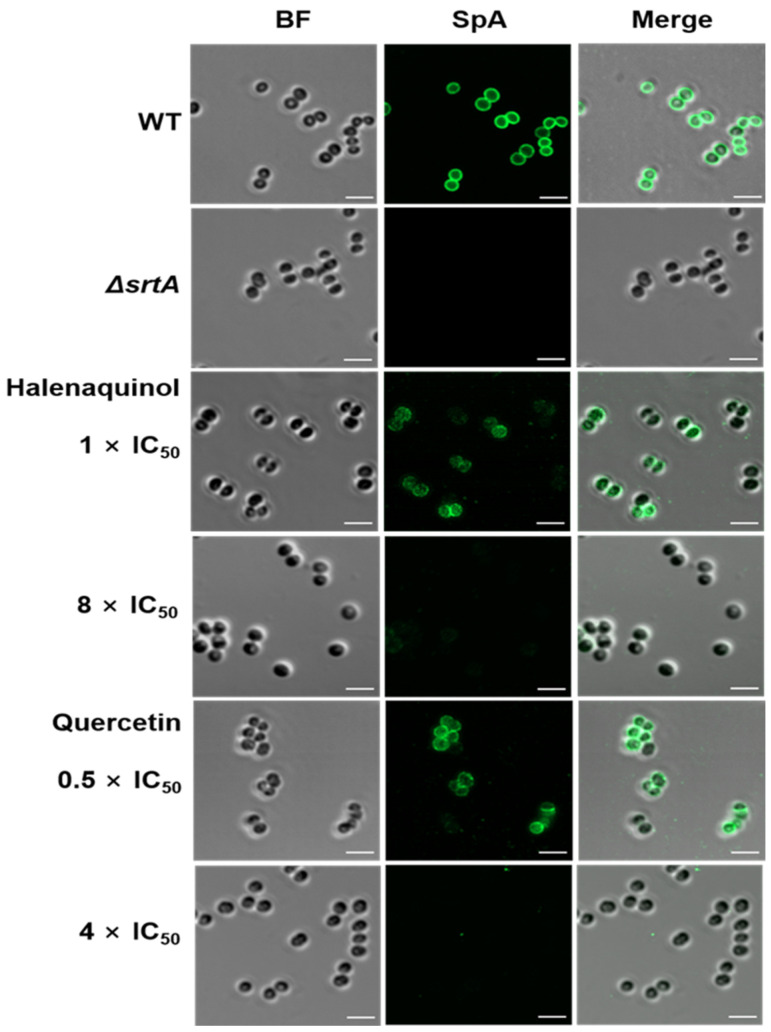
Fluorescence microscopy images of SpA surface display. Surface-displayed SpA of *S. aureus* Newman (wild-type; WT), isogenic *srtA*-knockout mutant (*ΔsrtA*), and wild-type treated with the indicated SrtA IC_50_ values of halenaquinol and quercetin (control) were immunostained with a fluorescence antibody and captured using confocal microscopy. BF identifies the bright-field microscopy view of fluorescence microscopy images. Scale bar, 2 µM.

**Table 1 marinedrugs-22-00266-t001:** Inhibitory effects of compounds **1**–**6** on SrtA activity and *S. aureus* ATCC6538p growth.

Compound	SrtA IC_50_ μM (μg/mL) **	MIC μM (μg/mL)
**1**	103.48 ± 8.74 (42.88 ± 3.62)	308.90 (>128)
**2**	13.94 ± 1.06 (4.66 ± 0.44)	382.88 (>128)
**3**	51.76 ± 6.69 (16.79 ± 2.17)	394.62 (>128)
**4**	>319.69 (>128)	319.69 (>128)
**5**	131.09 ± 19.44 (57.87 ± 8.58)	289.96 (>128)
**6**	47.52 ± 12.27 (15.41 ± 3.98)	394.67 (>128)
Quercetin *	157.62 ± 8.40 (47.64 ± 2.54)	>423.50 (>128)
Ampicillin *	ND ***	0.4 (0.1)

* Positive control. ** The values represent the mean ± SD (*n* = 3). *** Not determined.

## Data Availability

All data are contained within this article and the Appendix A.

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
