# Peer review of "Halenaquinol Blocks Staphylococcal Protein A Anchoring on Cell Wall Surface by Inhibiting Sortase A in Staphylococcus aureus"

_marinedrugs, 2024, doi:10.3390/md22060266_

Round 1

Reviewer 1 Report

Comments and Suggestions for Authors

The manuscript entitled "Halenaquinol Blocks Staphylococcal Protein A Anchoring on 2 Cell Wall Surface by Inhibiting Sortase A in Staphylococcus 3
aureus" is very well written and the study was well designed. However, there some small points I would like to suggest in order to improve the manuscript:
- The authors always wrote the compound number after the compound's name. It is quite annoying and confusing for readers because it somehow confuse me with references. I suggest the authors to only write the numbering once and delete the compound numbering throughout the manuscript. Writing the name is sufficient in this case.

- As the authors wrote in the discussion, it would be much better if the author could perform some bioinformatic analysis. I suggest the authors perform molecular docking to predict where the compounds bind to the enzyme whether they act as competitive or allosteric inhibitor.

- The authors need to describe how they determined the IC50 in the manuscript.

Comments on the Quality of English Language

The english quality of the manuscript is good enough. There are only some typos or grammatical errors that can be corrected easily.

Author Response

[Responses to Reviewer 1]

Thank you very much for your careful and valuable review of our manuscript. We made all revisions and corrections as far as we could. I hope this is the right answer for your request. What follows is our response to reviewer’s critique with the explanation of the changes implemented in the paper and a rebuttal when appropriate.

Comment 1:

The authors always wrote the compound number after the compound's name. It is quite annoying and confusing for readers because it somehow confuse me with references. I suggest the authors to only write the numbering once and delete the compound numbering throughout the manuscript. Writing the name is sufficient in this case.

Answer)

According to the reviewer’s comments, we revised and only write the numbering once in 2.1. Structural Elucidation of Compounds 1‒6 section (page 2, lines 82-90) and delete the compound numbering throughout the manuscript.

Comment 2:

As the authors wrote in the discussion, it would be much better if the author could perform some bioinformatic analysis. I suggest the authors perform molecular docking to predict where the compounds bind to the enzyme whether they act as competitive or allosteric inhibitor.

Answer)

We appreciate Reviewer’s kind comments. Currently, we are planning to analyze the action patterns of inhibitors and SrtA using molecular docking or bioinformatic analysis. Initially, in this paper, we performed kinetic studies on SrtA using halenaquinol and added the results (page 3, lines 119-124): To determine the type of inhibition, kinetic studies were performed with halenaquinol at IC50 or twofold IC50 based on a Lineweaver and Burk plot [27]. Inhibitor constants were obtained by a Dixon plot. Inhibitory kinetics show that halenaquinol behaved as a mixed inhibitor (Ki= 9.55 µM). Moreover, the binding of halenaquinol to SrtA was reversible because the enzyme activity was recovered by dialysis, excluding the possible existence of a covalent bond between inhibitor and enzyme.

Comment 3:

The authors need to describe how they determined the IC50 in the manuscript.

Answer)

We revised the manuscript and provided a detailed description of the IC50 determination procedures in 4.3. SrtA Inhibition Assay section (page 8, line 301- page 9, line 318). As described in this section, the percentage of SrtA inhibition was calculated using the following equation: {1 - (Difference value of the test compound / Difference value of a negative control)} × 100. A 12-point, twofold serial dilution dose–response assay was performed independently in triplicate. The resultant dose–response concentration range was 128 to 0.1 μg/mL of inhibitor. Data were scaled to internal controls, and a four-parameter logistic model (SigmaPlot ver. 10.0) was used to fit the measured data and determine the IC50 values.

Reviewer 2 Report

Comments and Suggestions for Authors

marinedrugs-3042541

Halenaquinol Blocks Staphylococcal Protein A Anchoring on Cell Wall Surface by Inhibiting Sortase A in Staphylococcus aureus

The article presents interesting research on pentacyclic polyketides as anti-infective agents targeting S. aureus.  The research is well developed, focusing on the compound’s interaction with sortase and also on their effect on bacteria.

The authors should present in table 1 the micromolar concentrations along the ones in mg/mL. Please add some statistical measure of error for the IC50 values. Present how many replicates were done, the number of concentrations tested and how IC50 was calculated (how was the inhibition curve drawn?).

I don’t really understand the use of quercetin as positive control. Most flavones have a very low solubility in water making their assays quite difficult. Another possible problem would be the intrinsic fluorescence of quercetin and also of the tested compounds. The authors should measure their fluorescence in the presence of SrtA, but not of the substrate. Usually, phenyl-vinyl-sulfone seem to be preferred as control in Sortase based assays. The authors should justify the use of quercetin 

In my view, the structure activity relationships should be done with the IC50 values expressed as micromolar units. The difference in molecular weight could be a major cause for the activity differences. As describe now, the SAR are not very exact. Compound 6 has a good SrtA inhibition even if the 3-carbonyl is missing. As the authors mentioned, a docking study or molecular dynamics are important to understand the mechanism of action.

It seems that the authors followed the reaction for 1 hour. Based on the reaction kinetics, was the inhibition profile reversible or irreversible?

The discussion section could be improved by discussion other inhibitors of SrtA. See and reference the following review presenting a large collection of SrtA inhibitors and their chemical classification.

“Targeting Bacterial Sortases in Search of Anti-Virulence Therapies with Low Risk of Resistance Development”, Pharmaceuticals. 2021 Apr 30;14(5):415.

Discuss the new compounds in the context of similar others. The authors did something like this, but it seems to be limited to their previous studies.

Comments on the Quality of English Language

needs small corrections.

Author Response

[Responses to Reviewer 2]

Thank you very much for your careful and valuable review of our manuscript. Your comments are encouraged us in doing our hardest scientific work in this research field. We made all revisions and corrections as far as we could. I hope this is the right answer for your request. What follows is our response to reviewer’s critique with the explanation of the changes implemented in the paper and a rebuttal when appropriate.

Comment 1:

The authors should present in table 1 the micromolar concentrations along the ones in mg/mL. Please add some statistical measure of error for the IC50 values. Present how many replicates were done, the number of concentrations tested and how IC50 was calculated (how was the inhibition curve drawn?).

Answer)

We appreciate Reviewer’s kind comments. We revised and provided the micromolar concentrations along the ones in mg/mL in Table 1, with statistical measure of error for the IC50 values obtained independently in triplicate (page 4). In addition, we provided a detailed description of the IC50 determination procedures in 4.3. SrtA Inhibition Assay section (page 8, line 301- page 9, line 318). As described in this section, the percentage of SrtA inhibition was calculated using the following equation: {1 - (Difference value of the test compound / Difference value of a negative control)} × 100. A 12-point, twofold serial dilution dose–response assay was performed independently in triplicate. The resultant dose–response concentration range was 128 to 0.1 μg/mL of inhibitor. Data were scaled to internal controls, and a four-parameter logistic model (SigmaPlot ver. 10.0) was used to fit the measured data and determine the IC50 values.

Comment 2:

I don’t really understand the use of quercetin as positive control. Most flavones have a very low solubility in water making their assays quite difficult. Another possible problem would be the intrinsic fluorescence of quercetin and also of the tested compounds. The authors should measure their fluorescence in the presence of SrtA, but not of the substrate. Usually, phenyl-vinyl-sulfone seem to be preferred as control in Sortase based assays. The authors should justify the use of quercetin 

Answer)

We appreciate Reviewer’s valuable comments and agree with points he made. In our experiences, many SrtA inhibitors also show srtA transcription inhibitory activity. However, quercetin did not inhibit srtA transcription like halenaquinol. Therefore, we used it as a positive control. Most flavons, even if dissolved in DMSO, are typically suspended due to their low solubility when diluted in water. In our experiment, quercetin and test compounds dissolved in DMSO as stock solutions and then diluted with SrtA reaction buffer (50 mM Tris-HCl pH 7.5, 150 mM NaCl, and 5 mM CaCl2) to yield a final DMSO concentration of 1%. In this case, quercetin dissolved well in reaction buffer. In addition, fluorescence of quercetin in the presence of SrtA, but not of the substrate, was not detected and was at a negative control (DMSO-treated) level using a microplate reader at excitation and emission wavelengths of 320 nm and 420 nm, respectively. We revised the manuscript and provided a detailed description of the IC50 determination procedures in 4.3. SrtA Inhibition Assay section (page 8, line 301- page 9, line 318).

Comment 3:

In my view, the structure activity relationships should be done with the IC50 values expressed as micromolar units. The difference in molecular weight could be a major cause for the activity differences. As describe now, the SAR are not very exact. Compound 6 has a good SrtA inhibition even if the 3-carbonyl is missing. As the authors mentioned, a docking study or molecular dynamics are important to understand the mechanism of action.

Answer)

As mentioned above, we revised and provided the micromolar concentrations along the ones in mg/mL in Table 1, with statistical measure of error for the IC50 values obtained independently in triplicate (page 4). As the reviewer commented, the SAR are not very exact. Compound 6, which has a hydroxyl group at the C-3 position, shows a good SrtA inhibition even if the 3-carbonyl is missing. Therefore, we revised the description as follows (page 3, lines 111-115): In contrast, xestolactone A, which has a hydroxyl group at the C-3 position, showed strong SrtA inhibitory activity (IC50 = 47.52 μM), indicating the crucial role of the carbonyl group or the hydroxyl group at the C-3 position in the SrtA activity inhibition of these compounds [According to the Reviewer 1’s comments, we revised and only write the numbering once in 2.1. Structural Elucidation of Compounds 1‒6 section (page 2, lines 82-90) and delete the compound numbering throughout the manuscript].

Comment 4:

It seems that the authors followed the reaction for 1 hour. Based on the reaction kinetics, was the inhibition profile reversible or irreversible?

Answer)

We performed kinetic studies on SrtA using halenaquinol and the results were provided in the revised version (page 3, lines 119-124): To determine the type of inhibition, kinetic studies were performed with halenaquinol at IC50 or twofold IC50 based on a Lineweaver and Burk plot [27]. Inhibitor constants were obtained by a Dixon plot. Inhibitory kinetics show that halenaquinol behaved as a mixed inhibitor (Ki= 9.55 µM). Moreover, the binding of halenaquinol to SrtA was reversible because the enzyme activity was recovered by dialysis, excluding the possible existence of a covalent bond between inhibitor and enzyme.

Comment 5:

The discussion section could be improved by discussion other inhibitors of SrtA. See and reference the following review presenting a large collection of SrtA inhibitors and their chemical classification. “Targeting Bacterial Sortases in Search of Anti-Virulence Therapies with Low Risk of Resistance Development”, Pharmaceuticals. 2021 Apr 30;14(5):415. Discuss the new compounds in the context of similar others. The authors did something like this, but it seems to be limited to their previous studies.

Answer)

We appreciate Reviewer’s valuable comments. As the reviewer commented, there are some recent reviews presenting a large collection of SrtA inhibitors. Although limited in explaining many examples, we tried to provide a better discussion for some of the interesting inhibitors of spongy origin, synthetic low molecules, and medicinal plant origin. We also cited related reviews including Pharmaceuticals 2021, 14, 415 in the revised version (page 7, lines 213-233).

Comment 6:

Comments on the Quality of English Language: needs small corrections.

Answer)

The English in this document has been checked by MDPI English editing (Journal submission Rapid, English editing ID: english-80839). For a certificate, please see: https://www.mdpi.com

Round 2

Reviewer 1 Report

Comments and Suggestions for Authors

I thank the authors for putting effort in revising and improving the manuscript. Some of the points I suggested have been addressed. However, one point about the bioinformatic analysis to reveal the mechanism of action of halequinol whether it works as competitive or allosteric antagonist has not been addressed. The authors argumented that the possible action of thes compound can be predicted by the Dixon plot and inhibitor kinetic studies. However, both of them are not provided in the manuscript. In my opinion, if the authors claim that the bioinformatic analysis is not necessary for now and it can be assumed from the kinetic studies, the authors have to provide these data/results in the main manuscript, both the kinetic studies and the relation to Dixon plot. The authors also need to describe it in method part.

Another point I would like to suggest is about tructure-activity relationship (SAR) analysis. The authors mentioned it in discussion and conclusion. Hence, no method and results/graph representing this is provided. I suggest that the authors need to provide the method of the tructure-activity relationship (SAR) analysis as well as the data in the manuscript.

Author Response

[Responses to Reviewer 1]

Thank you very much for your careful and valuable review of our manuscript. We made all revisions and corrections as far as we could. I hope this is the right answer for your request. What follows is our response to reviewer’s critique with the explanation of the changes implemented in the paper.

Comment 1:

I thank the authors for putting effort in revising and improving the manuscript. Some of the points I suggested have been addressed. However, one point about the bioinformatic analysis to reveal the mechanism of action of halequinol whether it works as competitive or allosteric antagonist has not been addressed. The authors argumented that the possible action of these compound can be predicted by the Dixon plot and inhibitor kinetic studies. However, both of them are not provided in the manuscript. In my opinion, if the authors claim that the bioinformatic analysis is not necessary for now and it can be assumed from the kinetic studies, the authors have to provide these data/results in the main manuscript, both the kinetic studies and the relation to Dixon plot. The authors also need to describe it in method part.

Answer)

We appreciate Reviewer’s valuable comments. The results of Dixon plot and inhibitor kinetic studies are provided in Figure 2 in the revised version, and the experimental procedures are described in 4.3. SrtA Inhibition Assay (page 10, lines 323-327) in method part.

Comment 2:

Another point I would like to suggest is about structure-activity relationship (SAR) analysis. The authors mentioned it in discussion and conclusion. Hence, no method and results/graph representing this is provided. I suggest that the authors need to provide the method of the structure-activity relationship (SAR) analysis as well as the data in the manuscript.

Answer)

As the reviewer pointed out, we did not conduct structure-activity relationship (SAR) analysis in this study. It seems to be a misrepresentation to express it as SAR. Therefore, we revised "The structure-activity relationship (SAR) analysis indicated" to "The SrtA inhibitory activity data of the isolated compounds indicated " in discussion and conclusion.

Reviewer 2 Report

Comments and Suggestions for Authors

The authors improved significantly their manuscript. I suggest only to include some of the comments athey provided to the review în the final paper. It would be intersting for the readers to better understand the assay nethodology. Otherwise, the manuscript can be accepted.

Comments on the Quality of English Language

Ok

Author Response

[Responses to Reviewer 2]

Comment 1:

The authors improved significantly their manuscript. I suggest only to include some of the comments as they provided to the review in the final paper. It would be interesting for the readers to better understand the assay methodology. Otherwise, the manuscript can be accepted.

Answer)

We appreciate Reviewer’s kind comments. Thank you very much for your careful and valuable review of our manuscript. Your comments are encouraged us in doing our hard scientific work in this research field. We made all revisions and corrections as far as we could.

Round 3

Reviewer 1 Report

Comments and Suggestions for Authors

I thank the authors for the effort in revising and improving the manuscript.